# MYCN Amplification Is Associated with Reduced Expression of Genes Encoding γ-Secretase Complex and NOTCH Signaling Components in Neuroblastoma

**DOI:** 10.3390/ijms24098141

**Published:** 2023-05-02

**Authors:** Prasoon Agarwal, Aleksandra Glowacka, Loay Mahmoud, Wesam Bazzar, Lars-Gunnar Larsson, Mohammad Alzrigat

**Affiliations:** 1National Bioinformatics Infrastructure Sweden (NBIS), Science for Life Laboratory, Division of Occupational and Environmental Medicine, Department of Laboratory Medicine, Lund University, 22362 Lund, Sweden; 2Department of Microbiology, Tumor and Cell Biology, Karolinska Institutet, 17165 Solna, Sweden; 3Department of Cell and Molecular Biology, Karolinska Institutet, 17177 Stockholm, Sweden; 4Department of Pharmaceutical Biosciences, Biomedical Center, Uppsala University, 75124 Uppsala, Sweden

**Keywords:** neuroblastoma, MYCN amplification, MYCN, γ-secretase, NOTCH signaling

## Abstract

Amplification of the *MYCN* oncogene is found in ~20% of neuroblastoma (NB) cases and correlates with high-risk disease and poor prognosis. Despite the plethora of studies describing the role of MYCN in NB, the exact molecular mechanisms underlying MYCN’s contribution to high-risk disease are not completely understood. Herein, we implemented an integrative approach combining publicly available RNA-Seq and MYCN ChIP-Seq datasets derived from human NB cell lines to define biological processes directly regulated by MYCN in NB. Our approach revealed that *MYCN-*amplified NB cell lines, when compared to non-*MYCN*-amplified cell lines, are characterized by reduced expression of genes involved in NOTCH receptor processing, axoneme assembly, and membrane protein proteolysis. More specifically, we found genes encoding members of the γ-secretase complex, which is known for its ability to liberate several intracellular signaling molecules from membrane-bound proteins such as NOTCH receptors, to be down-regulated in *MYCN-*amplified NB cell lines. Analysis of MYCN ChIP-Seq data revealed an enrichment of MYCN binding at the transcription start sites of genes encoding γ-secretase complex subunits. Notably, using publicly available gene expression data from NB primary tumors, we revealed that the expression of γ-secretase subunits encoding genes and other components of the NOTCH signaling pathway was also reduced in *MYCN-*amplified tumors and correlated with worse overall survival in NB patients. Genetic or pharmacological depletion of MYCN in NB cell lines induced the expression of γ-secretase genes and NOTCH-target genes. Chemical inhibition of γ-secretase activity dampened the expression of NOTCH-target genes upon MYCN depletion in NB cells. In conclusion, this study defines a set of MYCN-regulated pathways that are specific to *MYCN*-amplified NB tumors, and it suggests a novel role for MYCN in the suppression of genes of the γ-secretase complex, with an impact on the NOTCH-target gene expression in *MYCN-*amplified NB.

## 1. Introduction

Neuroblastoma (NB) is the most common extra-cranial solid tumor form during childhood, and it is believed to arise from the neural crest [1,2,3]. It has been estimated that NB accounts for approximately 15% of cancer-related deaths in children, with the majority of cases (90%) diagnosed by the age of five years [1,2,3]. NB patients are stratified into low-, intermediate-, and high-risk groups based on age, tumor histology, clinical stage, and genetic makeup [4,5,6,7]. Low- and intermediate-risk patients have favorable outcomes with an 80–95% event-free survival rate, while high-risk patients demonstrate a <50% event-free survival rate [1,2,3]. The current treatment of high-risk patients includes traditional intensive chemotherapy followed by surgical removal, radiation, myeloablation and autologous bone marrow transplantation, and more recently, immunotherapy [8,9,10]. However, the majority of high-risk patients relapse and eventually die due to refractory disease. Thus, high-risk NB is considered a disease with unmet medical need.

Amplification of the *MYCN* gene is detected in ~20% of all NB cases, and in about 40% of high-risk NB cases, and it is the genetic aberration most consistently associated with high-risk disease and poor survival [11,12]. MYCN belongs to the MYC family of oncoproteins that also includes MYC (c-MYC) and MYCL. The *MYC* genes encode transcription factors of the basic region/helix-loop-helix/leucine zipper (bHLHZip) family and control genes involved in multiple fundamental cellular processes, such as cell cycle progression, metabolism, apoptosis, senescence, differentiation, stem cell functions, and angiogenesis [13,14,15,16,17,18]. To regulate expression of their target genes, MYC family proteins need to dimerize with another bHLHZip protein called MAX, resulting in a stable heterodimer formation (hereafter called MYC:MAX) that binds and regulates MYC target genes [19,20]. MYC has been described as a general amplifier of transcription for all active genes in a cell [21,22,23,24], while later studies suggest that this effect is indirect, and is observed subsequently as a consequence of MYC’s regulation of specific genes [25,26,27]. In addition to target gene activation, MYC has also been reported to repress the transcription of genes, often in complexes with the zinc finger protein MIZ-1 [26,28,29]. The crucial role of MYC-family oncoproteins in cancer development makes them attractive targets for therapy in cancer, including NB. Therefore, several efforts have been put forward to identify molecules that dampen MYC activity in cancer [30,31,32,33,34]. Even though some of these molecules have demonstrated promising anti-tumor activity in vitro and in vivo, no specific MYC inhibitor is yet used in clinical practice. In addition to MYCN amplification, other genetic aberrations, such as TERT rearrangements or alternative lengthening of telomeres (ALT), also define a subset of high-risk NB with a negative poor outcome [35,36,37]. Furthermore, also other genetic aberrations, such as mutations in ALK, PTPN11, ATRX, TERT, and NRAS, have been shown to correlate with advanced-stage and high-risk NB [38,39].

Although *MYCN* amplification is correlated with high-risk and aggressive NB, the molecular mechanisms underpinning this association are still somewhat elusive. Herein, we analyzed publicly available gene expression datasets derived from 39 commonly used human NB cell lines representing *MYCN*-amplified and non-*MYCN*-amplified NB in an effort to identify pathways that are differentially regulated in *MYCN*-amplified NB tumors. On the one hand, we uncovered that *MYCN*-amplified cell lines exhibit increased expression of genes involved in ribosome biogenesis, RNA metabolism, gene expression, and protein synthesis processes. On the other hand, the *MYCN*-amplified cell lines were characterized by the down-regulated expression of genes related to NOTCH receptors and amyloid protein processing, axoneme assembly, protein proteolysis, localization, and transport. These findings were supported by an in-silico analysis of publicly available gene expression data derived from primary NB tumors, demonstrating the increased expression of genes involved in ribosomal biogenesis and the low expression of genes involved in NOTCH receptor processing and signaling in MYCN-amplified tumors when compared to non-*MYCN*-amplified tumors. Combining gene expression data with MYCN ChIP-Seq data in NB cell lines suggested the direct role of MYCN in regulating the expression of genes involved in the differentially regulated processes. Importantly, we identified MYCN as a direct or indirect repressor of members of the γ-secretase complex, which is known to activate signaling cascades downstream of membrane-bound proteins, such as NOTCH receptors. Consequently, down-regulation of MYCN through genetic or pharmacological means resulted in up-regulation of the repressed γ-secretase genes. In keeping with this, we demonstrated the increased expression of NOTCH-target genes upon MYCN depletion, which was dampened by the chemical inhibition of the γ-secretase complex in NB cell lines. These data provide new insights into the role of MYCN in NB, identify the genes and biological processes regulated by MYCN, and further suggest MYCN as a negative modulator of genes encoding the γ-secretase complex and NOTCH-target gene expression. The role of the NOTCH signaling pathway in NB is an active field of research, and this study provides further insights into the molecular mechanisms underlying NOTCH signaling regulation in NB.

## 2. Results

### 2.1. Identification of MYCN-Regulated Gene Expression Profiles and Biological Processes in MYCN-Amplified Neuroblastoma

To identify the biological processes that are specific to *MYCN*-amplified NB when compared to non-*MYCN*-amplified NB, we analyzed gene expression data derived from 39 commonly used human NB cell lines, as described by Harneza et al. [40], and we identified 2738 differentially regulated genes (FDR ≤ 0.05) in *MYCN-*amplified vs. non-*MYCN*-amplified NB (Appendix A). The gene ontology analysis of biological processes using the online Enricher analysis tool and the Molecular Signatures Database (MSigDB) uncovered that the up-regulated genes were related to ribosome biogenesis, ribosomal RNA synthesis and processing, RNA metabolism, protein translation, and gene expression (Figure 1A and Appendix A). The down-regulated genes were enriched in processes related to NOTCH receptor processing, protein proteolysis and localization, peptidase activity, and axoneme assembly (Figure 1B and Appendix A). To investigate whether similar correlations exist in primary tumors, we analyzed the gene expression of the SEQC study comprising RNA expression data from a 498-patient cohort with NB (r2.amc.nl/; Tumor Neuroblastoma-SEQC-498-custom-ag44kcwolf). As shown in Appendix A, the expression of genes linked to ribosomal RNA biogenesis (Appendix A) and the NOTCH receptor processing and signaling pathway (Appendix A) were, in general, higher and lower, respectively, in *MYCN*-amplified (green label) tumors when compared to non-*MYCN-*amplified (red label) tumors.

Next, we set out to uncover the potential direct regulation of the differentially regulated genes by MYCN and analyzed MYCN ChIP-Seq data in three *MYCN*-amplified human NB cell lines; KELLY, LAN-5, and COGN145 [41]. For the enrichment of the MYCN ChIP-Seq peaks, we defined ±5 kb genomic regions from known transcriptional start sites (TSSs) as a cut-off. Based on this definition, we identified 5870 annotated genes common to the three *MYCN-*amplified cell lines (Appendix A). Overlaying these common genes with the differentially expressed genes in the *MYCN*-amplified NB cell lines resulted in 475 up-regulated and 497 down-regulated genes, which we define as MYCN-target genes (Figure 1C). Figure 1D,E shows the average pileup of MYCN ChIP-Seq peaks around the TSSs of up- and down-regulated genes in the KELLY cell line and indicates that the MYCN binding sites are enriched around ±1 kb of the TSSs (Figure 1D,E). To explore the transcriptional regulation of genes differentially regulated by MYCN further, we utilized the MSigDB to define the transcription factor binding motifs. This analysis revealed an enrichment of the MYC-type of E-boxes and binding motifs of other transcription factors, such as E2F, NRF1, YY1, and NFY, at the TSSs of the MYCN-bound and up-regulated genes (Figure 1F top panel), while the TSSs of the MYCN-bound and down-regulated genes were mainly enriched with binding motifs for ETS family members, such as ELK1, ETS1, ETS2, GABPB, and TEL (Figure 1F, lower panel). We also found the enrichment of the NRF2, PAX4, and MYC-type E-box binding motifs at the TSSs of the MYCN-bound and down-regulated genes (Figure 1F, lower panel).

### 2.2. Integrative Analysis of Gene Expression and MYCN ChIP-Seq Reveals That MYCN Directly Associates with the Suppressed ADAM17 and γ-Secretase Complex Genes

Having defined the MYCN-bound and differentially regulated genes, we sought to uncover which of these genes are potentially directly regulated by MYCN. The gene ontology analysis revealed that the 475 up-regulated and MYCN-bound genes were related to RNA synthesis and processing, ribosome biogenesis, RNA metabolism, and protein translation (Figure 2A and Appendix A), while the 497 down-regulated MYCN-target genes were involved in NOTCH receptor processing, protein proteolysis and localization, endopeptidase activity, and amyloid protein metabolism (Figure 2A and Appendix A). A visualization of the MYCN ChIP-Seq peaks shows that they coincide with the TSSs of the up- and down-regulated genes identified in this study, such as *RPL5* and *APH1A*, respectively (Figure 2B, Appendix A). Intriguingly, we found the genes encoding for ADAM17 and the γ-secretase complex, which constitute the GO terms related to NOTCH receptor processing, amyloid protein processing, and membrane protein proteolysis, to be MYCN-bound and down-regulated in the *MYCN*-amplified cell lines (Table 1, Appendix A). Figure 2C,D shows the MYCN ChIP-Seq peak pileup around the TSSs of up-regulated genes involved in ribosomal biogenesis (Figure 2C) and the down-regulated ADAM17 and γ-secretase complex genes (Figure 2D) in the *MYCN*-amplified KELLY cell line [41], indicating that the MYCN binding sites are enriched around ±1 kb of TSSs.

Considering that MYC family proteins may affect the total RNA content per cell in certain cell types [21,22,23,24], and the fact that the publicly available RNA-Seq data from the study by Harneza et al. analyzed above did not contain spike-in control [40], we analyzed the expression levels of *ADAM17*, genes encoding the γ-secretase complex, as well as other components of NOTCH signaling in the publicly available gene expression dataset with spike-in RNA control [22]. This analysis revealed that *ADAM17* and all γ-secretase encoding genes, with the exception of *APH1B*, were less expressed in the *MYCN*-amplified neuroblastoma cell lines KELLY, SK-N-BE(2)C, and NGP compared with the non-*MYCN*-amplified SK-N-AS cell line (Appendix A). Moreover, the NOTCH-target genes *HES1*, *HES4*, and *HEYL* (Appendix A) appeared to be less expressed in the *MYCN*-amplified cell lines. Further, all the NOTCH receptor genes except *NOTCH4* demonstrated lower mRNA levels in the *MYCN*-amplified cell lines (Appendix A). In sum, these data indicate that *MYCN* amplification correlates with the reduced expression of genes encoding the γ-secretase complex, ADAM17, and genes that encode components of the NOTCH signaling pathway in *MYCN-*amplified NB cell lines.

To further document the role of MYCN in regulating the identified genes and processes, we analyzed publicly available gene expression data following the shRNA-mediated MYCN knockdown in the *MYCN*-amplified cell line IMR32 [42]. First, we identified 6677 differentially regulated genes (FDR ≤ 0.05), of which 3229 were up-regulated and 3448 were down-regulated following MYCN depletion (Appendix A). Pathway analysis of the top 10 biological processes revealed that the up-regulated genes were enriched in genes related to axon guidance, the neurotrophin signaling pathway, endocytosis, and autophagy, as well as the NOTCH signaling pathway (Appendix A). The down-regulated genes were related to DNA replication, the cell cycle, ribosome biogenesis, RNA transport, spliceosomes, and DNA repair (Appendix A). When overlapping differentially regulated genes following MYCN depletion in the *MYCN*-amplified IMR32 cell line with our defined MYCN targets based on MYCN ChIP-Seq, we identified 1004 up-regulated genes and 1623 down-regulated genes as MYCN targets in IMR32 cells (Appendix A). Notably, the genes identified in this study involved in NOTCH signaling, such as *APH1A*, *APH1B*, and *PSEN1*, were among the MYCN-bound genes belonging to the up-regulated genes in response to MYCN depletion in the IMR32 cell line (Appendix A), while the MYCN-bound genes related to DNA replication, the cell cycle, RNA splicing, ribosomal biogenesis, and DNA repair were down-regulated (Appendix A). Moreover, the use of more stringent conditions (FDR ≤ 0.05 combined with FC ≥ 1) identified 692 up-regulated genes and 606 down-regulated genes (Appendix A). A pathway analysis of these genes revealed the up-regulation of genes related to the NOTCH signaling pathway and axon guidance, while the down-regulated genes were involved in DNA replication and repair, as well as the cell cycle (Figure 2E). These data suggest that MYCN, in addition to its activation of growth-related genes, is a potential direct suppressor of genes that belong to the γ-secretase complex and other genes involved in the NOTCH pathway.

### 2.3. MYCN Amplification Correlates with Reduced Expression of Genes of the γ-Secretase Complex and ADAM17 in Neuroblastoma

To strengthen the analysis of our bioinformatic approach demonstrating the down-regulation of genes encoding γ-secretase and ADAM17 in the 39-human NB-cell-line dataset (Figure 3A and Appendix A), we sought to evaluate the baseline expression levels of genes encoding γ-secretase and ADAM17 in a set of human NB cell lines representative of *MYCN*-amplified and non-*MYCN*-amplified NB. First, we assessed the MYCN and c-MYC protein levels in the NB cell lines used in this study to functionally validate the gene expression levels. Western blot analysis showed that the *MYCN*-amplified NB cell lines IMR32, KELLY, SK-N-BE(2), and CHP212 expressed high levels of the MYCN protein, while the non-*MYCN*-amplified NB cell lines expressed either c-MYC (SH-SY5Y and SK-N-AS) or a low level of MYCN (SK-N-FI) relative to the other cell lines (Appendix A). Next, to validate the impact of *MYCN* amplification on the expression of genes encoding γ-secretase complex subunits and ADAM17, we analyzed the baseline mRNA levels of these genes in those human NB cell lines by quantitative real-time PCR (qRT-PCR). The transcript levels of genes encoding the γ-secretase complex subunits and ADAM17 were found to be significantly lower in the *MYCN*-amplified NB cell lines IMR32, KELLY, SK-N-BE(2), and CHP212 compared with the non-*MYCN*-amplified NB cell lines SK-N-FI, SH-SY5Y, and SK-N-AS (Figure 3B). To test whether the expression patterns of these genes are similar in primary human NB tumors as well, we analyzed their expression using the gene expression dataset from the Tumor Neuroblastoma—SEQC study (r2.amc.nl/; Tumor Neuroblastoma-SEQC-498-custom-ag44kcwolf). Like our gene expression analysis on human NB cell lines, we found the expression of genes encoding members of the γ-secretase complex and ADAM17 to be significantly lower in *MYCN*-amplified primary NB tumors when compared with non-*MYCN*-amplified tumors (Figure 3C). Notably, the low expressions of *NCSTN*, *APH1B*, and *PSEN1* encoding for members of the γ-secretase complex correlated with poor overall survival in the analyzed cohort of 498 NB patients (Figure 3D), while other members were not indicative of any changes in overall survival.

### 2.4. Genetic and Pharmacological Depletion of MYCN Increases mRNA Levels of γ-Secretase Subunits and ADAM17 in Neuroblastoma

Having identified MYCN as a potential direct suppressor of the expression of ADAM17 and γ-secretase complex encoding genes, we sought to test whether MYCN genetic or pharmacological depletion in NB would increase their expression. To this end, we utilized the TET21N NB cell line, in which we could regulate the MYCN expression by doxycycline. As shown in Figure 4A, treatment of the TET21N cell line with 1 µg/mL doxycycline for 72 h completely depleted the MYCN protein levels (Figure 4A). Doxycycline-mediated MYCN depletion in the TET21N cells reduced cell proliferation and metabolic activity without affecting cell viability (Appendix A). Analysis of the mRNA levels of genes encoding the γ-secretase complex and ADAM17 by RT-qPCR revealed a significant increase in the mRNA levels following doxycycline treatment (Figure 4B). Next, we analyzed the impact of MYCN depletion using the newly described MYC:MAX inhibitor MYCMI-7 [43,44]. Utilizing the in situ proximity ligation assay (isPLA), we showed that 5 µM of MYCMI-7 inhibited endogenous MYCN:MAX interactions in the *MYCN*-amplified KELLY cell line 5 h post-treatment (Figure 4C,D). Treatment for 72 h with 0.4 µM MYCMI-7 reduced MYCN protein levels in the *MYCN-*amplified NB cell line KELLY (Figure 5E). Moreover, MYCMI-7 reduced cell proliferation and viability and induced cell death, as shown by the increased levels of cleaved-PARP protein in the KELLY cell line (Appendix A). Notably, MYCMI-7 treatment induced the expression of the γ-secretase complex encoding genes, as well as ADAM17, in the *MYCN*-amplified KELLY cell line (Figure 4F). To exclude effects of the MYCN status on the total RNA synthesis per cell, we measured the total RNA concentration from equal numbers of MYCN-proficient or -deficient cells. As shown in Appendix A, the total RNA amount did not differ between the MYCN-expressing and MYCN-depleted TET21N in response to doxycycline treatment or as a consequence of the MYCMI-7 treatment of KELLY cells (Appendix A). These data indicate that MYCN depletion induces the expression of the γ-secretase complex and ADAM17 in *MYCN-*amplified NB cells, and they further substantiate our bioinformatic analysis suggesting that MYCN plays a role as a direct negative regulator of genes encoding the γ-secretase complex and ADAM17.

### 2.5. MYCN Depletion Induces Expression of NOTCH Target Genes in Neuroblastoma

Our analysis of gene expression datasets derived from human NB cell lines and primary tumors suggested that several components of the NOTCH pathway are the down-regulated in *MYCN*-amplified tumors. Thus, we investigated the impact of MYCN amplification on the expression of some canonical NOTCH target genes of the bHLH (basic-helix-loop-helix) family, such as *HES* (Hairy/enhancer of Split) and *HEY* (Hairy/Enhancer of Split related with YRPW motif) [45,46]. Our analysis of RNA-Seq data derived from the human NB cell lines uncovered significantly reduced expression of the NOTCH target genes *HES1*, *HES4*, *HEY1*, and *HEYL* in the *MYCN*-amplified cell lines (Figure 5A and Appendix A). In primary NB tumors, we also found that *HES1*, *HES4*, *HEY1* and *HEYL* were down-regulated in *MYCN*-amplified tumors when compared to non-*MYCN*-amplified tumors (Figure 5B). Notably, the low expression of these NOTCH target genes correlated with poor overall survival in NB patients (Figure 5C). To assess the impact of MYCN depletion on the expression of NOTCH target genes, we evaluated the expression of *HES1*, *HES4*, *HEY1*, and *HEYL* in response to the genetic or pharmacological depletion of MYCN. First, we evaluated the mRNA levels of these genes in the TET21N cell line following MYCN depletion in response to doxycycline treatment. Treatment of the TET21N cells with 1 µg/mL doxycycline for 72 h led to significant increases in the mRNA levels of *HES1*, *HES4*, and *HEY1* (Figure 5D). We were unable to detect any *HEYL* mRNA signals in the TET21N cell line. To further corroborate these findings, we analyzed the mRNA levels of these four genes in response to 72 h treatment with 0.4 µM MYCMI-7 in the KELLY cell line. Similarly, the MYCN pharmacological depletion by MYCMI-7 induced significant increases in the *HES1*, *HES4*, *HEY1* and *HEYL* mRNA levels when compared to the DMSO control treatment (Figure 5E).

To obtain better insights into the expression of the NOTCH signaling pathway components in NB, we also analyzed the mRNA levels of all the NOTCH receptors (NOTCH1, NOTCH2, NOTCH3, and NOTCH4) in the 39-human NB-cell-line dataset. As shown in Appendix A, we did not find significant differences in the NOTCH receptor mRNA levels between the *MYCN*-amplified and non-*MYCN*-amplified cell lines (Appendix A). However, NOTCH receptors 1–3 were less expressed in the *MYCN*-amplified neuroblastoma cell lines KELLY, SK-N-BE(2)C, and NGP compared with the non-*MYCN*-amplified SK-N-AS cell line in another gene expression dataset with spike-in RNA control (Appendix A). All NOTCH receptors were significantly lower in expression in primary tumors with *MYCN* amplification when compared to non-*MYCN*-amplified tumors (Appendix A). Notably, the low expressions of the NOTCH receptors were indicative of poor overall survival in NB patients (Appendix A). In sum, these data indicate that *MYCN*-amplified NB cell lines and primary tumors are characterized by the low expression of NOTCH target genes, and that the inhibition of MYCN activates their expression.

### 2.6. Inhibition of γ-Secretase Complex Abolishes Induced Expression of NOTCH Target Genes upon MYCN Depletion in Neuroblastoma

Having shown that MYCN depletion induces the expression of γ-secretase complex and NOTCH target genes, we sought to investigate whether the chemical inhibition of the γ-secretase complex abolishes the induced expression of the NOTCH target genes in response to MYCN depletion. We utilized N-[N-(3,5-difluorophenacetyl)-l-alanyl]-S-phenylglycine t-butyl ester (DAPT), which is an inhibitor of γ-secretase that blocks the proteolytic processing and release of the NICD [47,48]. To this end, we treated the TET21N cell line with 1 µg/mL doxycycline, 5 µM DAPT, or their combination for 72 h, and then analyzed the mRNA expression levels of the NOTCH target genes *HES1*, *HES4*, and *HEY1*. Similar to our earlier observations, the doxycycline-mediated depletion of MYCN significantly increased the expression of *HES1*, *HES4*, and *HEY1* when compared to the DMSO control treatment (Figure 6). DAPT treatment alone did not affect the mRNA levels of *HES1* and *HES4* but reduced *HEY1* mRNA levels when compared to the DMSO control treatment (Figure 6). Notably, DAPT treatment in combination with doxycycline abrogated the increases in *HES1*, *HES4*, and *HEY1* mRNA levels in response to MYCN depletion, resulting in mRNA expression levels of these genes that are comparable to the DMSO control levels (Figure 6). In conclusion, these findings suggest that the up-regulation of NOTCH target genes is mediated by the increased expression of ADAM17 and γ-secretase upon the biological depletion of MYCN.

## 3. Discussion

High-risk neuroblastoma (NB) is a fatal pediatric cancer that is characterized by aggressive and undifferentiated tumors, drug resistance, and poor prognosis. Therefore, better understanding the molecular mechanisms underlying high-risk NB tumor development, progression, and resistance to therapy is a necessity for improved diagnosis, management, and treatment of this deadly tumor [49,50,51]. Amplification of the *MYCN* oncogene is a well-established marker of poor prognosis and high-risk NB [11,12]. MYCN, as well as the other MYC family of oncoproteins, function as central hubs for almost all signaling pathways; thus, they modulate and integrate a broad range of pathways involved in fundamental biological processes, such as cell proliferation, cell death, pluripotency, differentiation, senescence, cellular plasticity, cellular energetics, and metabolism [52,53,54]. Since the discovery of *MYCN* amplification in NB, extensive research efforts have been undertaken to uncover the role of MYCN in NB and its contribution to high-risk tumors. These efforts have connected deregulated MYCN expression to all hallmarks of NB, such as proliferation, blocked differentiation, metastatic spread, stemness, metabolic re-wiring, the suppression of immunosurveillance mechanisms, and the emergence of drug resistance [55,56,57,58,59]. However, the direct correlation or causation between MYCN and the NB-relevant biological pathways in *MYCN-*amplified tumors has not been completely deciphered.

In this study, we implemented an integrative approach using publicly available gene expression datasets derived from large numbers of human NB cell lines to define the biological pathways that are specific to *MYCN-*amplified tumors when compared to non-*MYCN-*amplified tumors. This approach identified ribosomal biogenesis, ribosomal RNA synthesis and processing, as well as protein synthesis as the enriched pathways in *MYCN-*amplified cell lines. Supporting this, we also found a positive correlation between these pathways and MYCN amplification in the RNA expression data of primary NB tumors. This is in line with previous studies demonstrating the dependency of *MYCN-*amplified tumors on these pathways. Based on MYCN overexpression studies on NB cell lines, Boon and colleagues [60] demonstrated that MYCN drives the expression of a large set of genes involved in ribosome biogenesis and protein synthesis [60]. Similarly, a strong correlation between MYCN expression and ribosome biogenesis has been documented in neuroblastoma patients, with a negative impact on patient survival [61]. It has been suggested that the factors involved in ribosomal biogenesis and ribosomal RNA processing are essential factors in the pathogenesis of *MYCN-*amplified NB tumors, and their genetic depletion demonstrates anti-NB effects [62,63,64,65]. We demonstrate that shRNA-mediated knockdown of *MYCN* in the *MYCN*-amplified cell line IMR32 resulted in down-regulated expression levels of genes related to RNA processing and ribosomal RNA synthesis and biogenesis, many of which were defined as MYCN-target genes in this study. By integrating MYCN ChIP-Seq data, our study suggests the direct regulation of genes involved in ribosomal biogenesis and RNA synthesis, as well as protein synthesis, by MYCN in NB. Furthermore, there was an enrichment of the MYC-type of E-box motifs around the TSSs of the up-regulated genes, further supporting the potential direct regulation of these genes by MYCN, and in agreement with previous reports [26,66]. Moreover, our analysis suggests that targeting ribosomal biogenesis could be a promising therapeutic strategy in high-risk NB with *MYCN* amplification. Indeed, blocking ribosome RNA expression using inhibitors of RNA polymerase I has demonstrated potent anti-NB activities in vitro and in vivo [61,67], which warrants further investigation and evaluation in clinical trials.

Our bioinformatic analysis of gene expression datasets revealed down-regulated expression of genes involved in the NOTCH signaling pathway in *MYCN-*amplified cell lines and primary tumors. Furthermore, our MYCN ChIP-Seq analysis revealed that genes encoding ADAM17 and subunits of the γ-secretase complex are direct MYCN target genes. Thus, this study focused on evaluating the impact of MYCN on the expression of *ADAM17* and the γ-secretase complex genes, as well as the expression of some NOTCH target genes in NB. It has been shown that NB cells express variable levels of NOTCH receptors, but low levels of NOTCH target genes [68], suggesting impaired NOTCH pathway activity in NB. Notably, Van Limpt, V.A., et al. [69] analyzed the expression of 21 genes involved in the NOTCH pathway in a large panel of NB cell lines, and the gene expression appeared to be the highest in the non-*MYCN-*amplified cell lines, such as SK-N-FI, LAN-6, SHEP, and LAN-2 [69]. Our findings demonstrating low expression of NOTCH receptors, ADAM17, γ-secretase, and NOTCH target genes in *MYCN-*amplified tumors, suggest that MYCN amplification contributes to the down-regulation of genes related to the NOTCH pathway in NB. This notion is supported by the demonstration that MYCN depletion induces the expression of genes encoding ADAM17, γ-secretase, and some NOTCH target genes. Further, we uncovered that the TSSs of ADAM17 and γ-secretase encoding genes are demarcated by MYCN, suggesting that MYCN is directly involved in the repression of these genes. It has been suggested that MYC/MYCN-mediated gene repression is dependent on interactions with MIZ-1 [26,29,70,71], epigenetic factors such as EZH2 histone methyltransferase and polycomb repressive complexes [71,72,73,74], histone deacetylase complexes [71,75], and DNA methyltransferase complexes [76,77]. Whether such mechanisms contribute to the reduced expression of genes encoding ADAM17, γ-secretase, and some NOTCH targets in *MYCN-*amplified tumors warrants further investigations. Studies in medulloblastoma have indicated that MYCN binds MIZ-1 with a lower affinity compared with MYC, with consequences for MIZ-dependent repression in different subtypes of medulloblastoma [78]. Possibly, MYCN-mediated transcriptional repression is therefore less dependent on MIZ-1. Interestingly, we found that MYCN-bound, repressed genes were enriched with DNA-binding motifs belonging to the ETS family of transcription factors, and, to a lesser extent, with the MYC-type of E-box motifs, compared with activated genes, suggesting that MYCN and ETS-family proteins may cooperate in the repression of transcription in neuroblastoma. [26,27,66]. Notably, The ETS family member ELK4 has been described as one of the factors that may drive the MYCN-mediated repression of gene transcription in NB [66]. Another ETS family member, ELK1, has been shown to negatively regulate the expression of some metalloproteases and members of the γ-secretase complex, such as PSEN1, PSEN2, and APH1A, when overexpressed in NB cells [79]. Furthermore, the γ-secretase complex is known for its wide range of substrates that act in many signal pathways [80,81]. Thus, it would be interesting to evaluate the impact of ADAM17 and the γ-secretase complex on substrates other than NOTCH receptors in *MYCN-*amplified tumors, and to define their biological contribution in the pathogenesis of such tumors. In addition, inhibitors targeting the γ-secretase complex have been suggested as potential anti-cancer agents [47,82,83]. The fact that our study demonstrates the low gene expression of the γ-secretase complex may argue against the therapeutic potential of γ-secretase inhibitors in *MYCN-*amplified NB tumors, which indeed demands further investigation.

It has been suggested that the NOTCH signaling pathway plays both oncogenic [84,85,86] and tumor suppressor [68] roles in NB. It is worth mentioning that reports describing the oncogenic functions of NOTCH signaling in NB are based on overexpression experiments, the use of chemical inhibitors, and in most cases, the utilization of non-*MYCN*-amplified cell lines, such as SH-SY5Y, which expresses c-MYC [84,85,86]. Cooperation between c-MYC and NOTCH during tumorigenesis has been documented in lung cancer [87], T-cell acute lymphocytic leukemia [88], and mantle cell lymphoma and chronic lymphocytic leukemia [89]. Whether NOTCH-c-MYC oncogenic cooperation occurs in non-*MYCN*-amplified NB with c-MYC expression demands further investigation. Recent studies investigating NB heterogeneity and cell identity based on core regulatory circuits (CRCs) and super enhancers (SEs) divided human NB cell lines and primary tumors into two main categories: undifferentiated mesenchymal neural crest-like NB (MES) and committed adrenergic cells (ADRN) [90,91]. Notably, *MYCN*-amplified cell lines, except for the CHP212 cell line, have demonstrated an enrichment of ADRN identity, while non-*MYCN*-amplified cell lines represented both identities [91]. A CRC and SE analysis of MES and ADRN in NB cell lines demonstrated an enrichment of NOTCH pathway genes in the MES but not the ADRN identity cell lines [92]. Even though MYC family members share common target genes and regulate similar pathways, and high c-MYC levels may drive high-risk NB [93], distinct transcriptional activities of MYCN and c-MYC have been suggested in NB [94]. These findings as well as our study demonstrating the reduced expression of genes involved in the NOTCH pathway in *MYCN*-amplified cell lines and primary tumors, point toward a possible antagonistic relationship between MYCN and the NOTCH pathway components in these NB tumors, but not in the non-*MYCN*-amplified NB expressing MYC. Intriguingly, high MYC expression in group 3 medulloblastoma has been shown to inversely correlate with the NOTCH signaling signature and reflects early fetal cerebellar development [95]. Therefore, the impact of NOTCH signaling on NB biology needs to be addressed carefully, as the biological outcomes of NOTCH signaling might be different depending on the subtype of NB (*MYCN*-amplification status), different experimental settings (physiological expression levels vs. overexpression), developmental stage and cell of origin, as well as the contribution of the tumor microenvironment (ligand activation vs. overexpression of NICD).

In summary, our study uncovers novel pathways directly regulated by MYCN in NB, and MYCN amplification correlates with the reduced expression of genes involved in the NOTCH signaling pathway compared to non-*MYCN*-amplified tumors. We also suggest that MYCN may function as a direct regulator dampening gene transcription of *ADAM17* and the γ-secretase complex genes in NB. However, further studies are required for a deeper understanding of the interplay between MYCN and the components of the NOTCH signaling pathway in NB.

## 4. Materials and Methods

### 4.1. RNA-Seq, ChIP-Seq Data, and Gene Ontology (GO) Analysis

The RNA-Sequencing (RNA-Seq) data of the 39 neuroblastoma cell lines under the accession number GSE89413 were downloaded from the gene expression Omnibus (GEO) database. Differentially expressed genes in *MYCN-*amplified vs. non-*MYCN-*amplified cell lines were analyzed using R scripts available at https://github.com/marislab/NBL-cell-line-RNA-seq, accessed on 23 February 2020, as described previously [40]. The raw sra files were downloaded using NCBI sratools. The reads were aligned using STAR version 2.7.0a [96]. The hg19 genome was used to align the reads. The aligned BAM files were then used to make the Tag files using the HOMER v4 module makeTagDirectory [97]. The HOMER module analyzeRepeats.pl was further used to generate the raw count file, which was used as an input to the R script provided in the original manuscript [40]. Data normalization and differential analysis were performed using the tool DESeq2 v 1.38.3 [98]. An adjusted *p*-value ≤ 0.05 was used as a final stringent measure to define the differentially expressed genes. MYCN ChIP-Seq data for the KELLY, COGN415, and LAN5 cell lines were downloaded from GEO under the accession numbers GSE94782 and GSE138295 [41]. All raw files and input data can be found at the Sequence Read Archive under the SRA accessions SRP223942. The downloaded data were aligned using BWA version0.7.17-r1188, and the human genome assembly hg19 was used as a reference. MACS peak-caller was used to call the peaks using the default parameters [99]. The peaks were annotated using the online tool GREAT v4.04 [100]. A cutoff of ±5 kb of the transcription start site (TSS) was used to define genes as MYCN-target genes. The molecular signatures database (MSigDB) and Enrichr online gene expression analysis tool [101,102,103] were utilized to investigate the gene ontologies of the differentially regulated genes in the 39-cell-line RNA-Seq data, and MYCN-bound genes were defined. The top 10 differentially regulated biological processes with a false discovery rate (FDR) ≤ 0.05 are presented in this study.

### 4.2. R2: Genomics Analysis and Visualization Platform

The R2: Genomics Analysis and Visualization Platform (http://r2.amc.nl, accessed on 15 March 2022) was used to analyze the expression of the NOTCH signaling pathway genes in primary NB tumors with or without MYCN amplification in the SEQC (GSE49710) dataset, and for the analysis of the spike-in gene expression arrays of the KELLY, SK-N-BE(2)c, NGP, and SK-N-AS cell lines (GSE80149). The R2 genome browser platform was utilized to show the MYCN enrichment at the transcription start sites of selected MYCN-regulated genes relevant to the findings in this paper. Statistical analysis was performed based on the R2: Genomics Analysis and Visualization Platform recommended setting, with an FDR ≤ 0.05 as the statistical significance threshold.

### 4.3. Cell Lines and Cell Cultures

SK-N-FI (ATCC catalog no. CRL-2142), SK-N-AS (ATCC catalog no. CRL-2137), KELLY (DSMZ catalog no. ACC-355), and SK-N-BE(2) (ATCC catalog no. CRL-2271) were kindly provided by Professor Per Kogner at the Karolinska Institutet. The IMR32 (ATCC catalog no. CCL-127), SH-SY5Y (ATCC catalog no. CRL-2266), and CHP-212 (ATCC catalog no. CRL-2273) cell lines were kindly provided by Professor Marie Arsenian Henriksson at the Karolinska Institutet. The TET21N cell line was obtained as a gift from Werner Lutz [104]. The *MYCN-*amplified cell lines IMR32, KELLY, SK-N-BE(2), and CHP212, as well as the TET12N cell line with doxycycline-regulatable MYCN expression, were maintained in RPMI 1640 medium (Gibco, ThermoFisher Scientific, Waltham, MA, USA). The SK-N-FI, SK-N-AS, and SH-SY5Y non-*MYCN-*amplified cell lines were maintained in DMEM medium (ThermoFisher Scientific, Waltham, MA, USA). The cell culture media contained GlutaMAX and were supplemented with 10% fetal bovine serum (ThermoFisher Scientific, Waltham, MA, USA) and 1% penicillin/streptomycin (ThermoFisher Scientific, Waltham, MA, USA). The IMR32 cell line culture was supplemented with nonessential amino acids (ThermoFisher Scientific, Waltham, MA, USA). Cell lines were grown in a humidifier incubator at 37 °C in 5% CO_2_. Treatment of cells with inhibitors or doxycycline was performed for 72 h. In brief, ≥90% viable cells were seeded in 10 cm dishes overnight and then treated with the indicated concentrations for 72 h. DMSO was used as the control treatment. All human neuroblastoma cell lines were STR-authenticated by short tandem repeat (STR) analysis (Eurofins Genomics, Ebersberg, Germany) once acquired by our laboratory. All cell lines were routinely tested for mycoplasma using the MycoAlert Mycoplasma Detection Kit (Lonza, Basel, Switzerland).

### 4.4. Resazurin Assay and Assessment of Cell Proliferation and Viability

For the resazurin assay assessment of the metabolic activity, the TET12N and KELLY cell lines were seeded in 96-well plates at a 3000 cells/well cell density and incubated overnight in a humidifier incubator at 37 °C in 5% CO_2_. The TET21N cell line was treated with 1 µg/mL doxycycline, while the KELLY cell line was treated with 0.4 µM MYCMI-7 for 72 h. DMSO was used as the control treatment. Then, resazurin reagent (Sigma, St. Louis, MO, USA) was added at a 44 µM final concentration. Reduced resazurin was detected by measuring the absorbance at 570 nm using a 96-well plate reader. A difference of 10× between the blank and DMSO-control-treated cells was set as the threshold. The cell proliferation was evaluated by counting the cell numbers over the treatment period, and the viability was assessed at the same time using trypan blue staining.

### 4.5. RNA Extraction, cDNA Synthesis, and RT-qPCR

RNA was isolated using the TRIzol extraction method following the manufacturer protocol (ThermoFisher Scientific, Waltham, MA, USA). The RNA concentration was measured using a Nanodrop, and 1 µg of RNA was used for cDNA synthesis using iScript™ Reverse Transcription Supermix (BioRad, Hercules, CA, USA), following the manufacturer protocol. Real-time quantitative PCR (RT-qPCR) was performed using iTaq™ Universal SYBR^®^ Green Supermix (BioRad, Hercules, CA, USA) in a 384-well PCR plate format. RT-qPCR reactions were performed using the C1000 Touch™ Thermal Cycler with the CFX384™ Real-time System (BioRad, Hercules, CA, USA). Analysis of the gene expression was performed using Bio-Rad CFX Maestro™ data analysis software version 1.1 (BioRad, Hercules, CA, USA) following the 2^−∆∆Ct^ method. HPRT1 and GAPDH were used as the housekeeping genes for the normalization of the RT-qPCR analysis. The primers used in this study are listed in Appendix A.

### 4.6. Protein Extraction and Western Blot

Cell lines were collected and washed with 1× cold phosphate-buffered saline (PBS). Cells were lysed with radioimmunoprecipitation assay RIPA buffer (Thermo Fisher Scientific, Waltham, MA, USA) supplemented with a protease and phosphatase inhibitor cocktail (Thermo Fisher Scientific) for 30 min on ice, sonicated (amplitude: 80; three cycles at 30 s each), and centrifuged (12,000× *g* for 15 min, 4 °C). The protein concentration was determined with a Pierce BCA Protein Assay (Thermo Fisher Scientific, Waltham, MA, USA). Equal amounts of protein (20 μg per sample) were separated on 4–12% Bis–Tris gradient gels (Thermo Fisher Scientific, Waltham, MA, USA) and blotted on a 0.2 µm nitrocellulose membrane (Bio-Rad, Hercules, CA, USA) with the Trans-Blot Turbo Transfer System (Bio-Rad, Hercules, CA, USA). Membranes were blocked for 1 h at room temperature, in 5% milk or bovine serum albumin (BSA) in PBS-Tween20 (PBS-T), followed by an overnight incubation with primary antibody at 4 °C. The following primary antibodies were used: anti-β-actin (A5441, Sigma-Aldrich, St. Louis, MO, USA) and anti-MYCN (sc-53993, Santa Cruz, CA, USA). Membranes were washed with PBS-T and incubated with secondary horseradish peroxidase-conjugated anti-mouse (ab97046, Abcam, Cambridge, UK) or anti-rabbit (ab97080, Abcam) antibody for 1 h at room temperature. To visualize the proteins, membranes were developed with Western Enhanced Chemiluminescence Substrates (Bio-Rad, Hercules, CA, USA). The ChemiDoc XRS+ System (Bio-Rad, Hercules, CA, USA) was used to determine the band intensity, along with Image Lab Software 5.1 for analysis.

### 4.7. In Situ Proximity Ligation Assay (isPLA)

The in-situ proximity ligation assay (isPLA) was performed as described previously elsewhere [37,80]. In brief, the *MYCN-*amplified KELLY cell line was used to detect endogenous MYCN:MAX interactions. KELLY cells were seeded in 96-well plates for 48 h, then treated with 5 µM of MYCMI-7 or DMSO for 5 h, and then washed twice with PBS and fixed with 4% paraformaldehyde for 10 min at room temperature. Cells were permeabilized with PBS with 0.05% Triton-X and incubated in a blocking buffer for 1 h at 37 °C, after which the isPLA was performed using the NaveniFlex MR kit following the manufacturer protocol (Navinci Diagnostics, Uppsala, Sweden). The following antibodies were used: mouse monoclonal anti-MYCN (sc-53993, Santa Cruz, CA, USA) and rabbit polyclonal anti-MAX (Abcam, catalog no. ab101271). Cells were stained with phalloidin and DAPI, and the isPLA signals were developed using Buffer C provided by the isPLA kit with Atto 647N. Image acquisition was performed using the ZEISS LSM 980 with an Airyscan 2 confocal microscope (Carl Zeiss Microscopy GmbH, Munich, Germany). All image stacks were acquired with comparable settings, at a resolution of 1024 × 1024 pixels and a z-stack size of 3 μm. The isPLA signals were quantified using CellProfiler 4.2.1 software. Data were analyzed using Rstudio version 4.1.0 with the following packages: tidyverse and ggplot2.

### 4.8. Statistical Analysis

Statistical analysis was performed using a two-tailed, unpaired Student *t*-test for the RT-qPCR data, and a one-way ANOVA for a comparison of the gene expression levels between the *MYCN-*amplified and non-*MYCN-*amplified cell lines and primary tumors, as well as for the isPLA data. Bonferroni correction of raw *p*-values was used for the Kaplan–Meier-analysis overall survival curve.

## Figures and Tables

**Figure 1 ijms-24-08141-f001:**
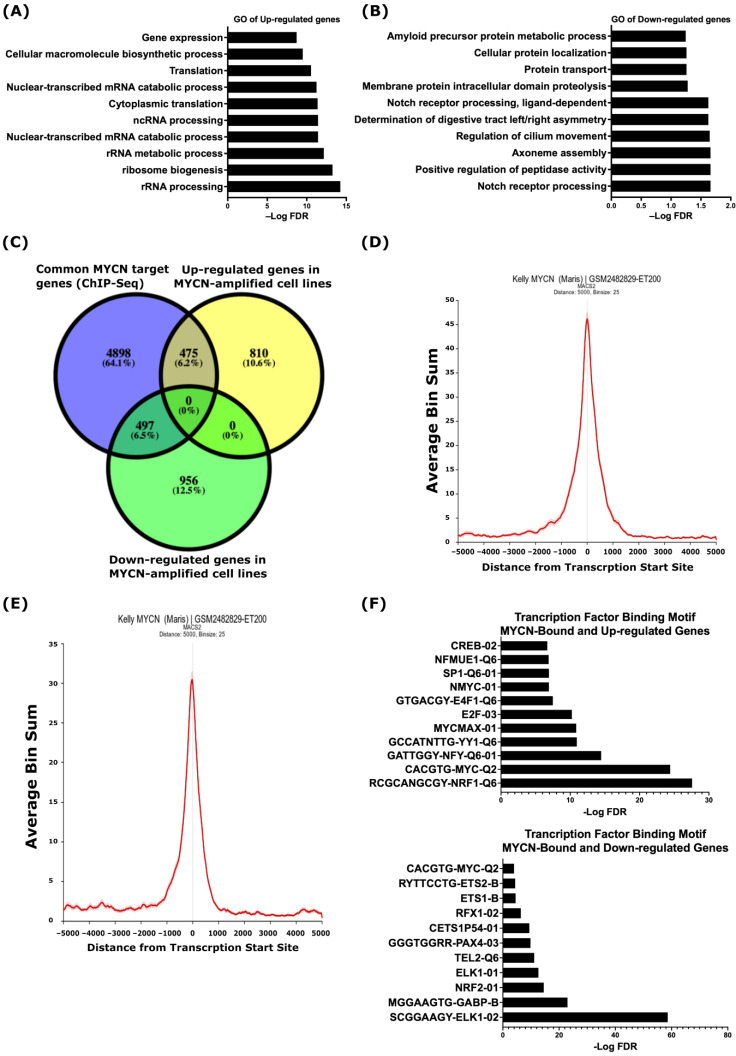
MYCN-regulated gene ontology terms in NB. Top 10 biological processes based on analysis of up-regulated (**A**) and down-regulated (**B**) genes in human NB *MYCN*-amplified cell lines when compared to non-*MYCN*-amplified cell lines. (**C**) Overlap between common MYCN target genes identified in three human *MYCN*-amplified cell lines and differentially regulated genes in *MYCN*-amplified human NB cell lines. Enrichment of MYCN ChIP-Seq peaks at the TSSs of up-regulated (**D**) and down-regulated (**E**) genes in the *MYCN*-amplified KELLY cell line. (**F**) Analysis of transcription factor binding motif. (**Top**), MYCN-bound and up-regulated genes. (**Bottom**), MYCN-bound and down-regulated genes. A false discovery rate (FDR) ≤ 0.05 was used as the cut-off value to define the statistical significance.

**Figure 2 ijms-24-08141-f002:**
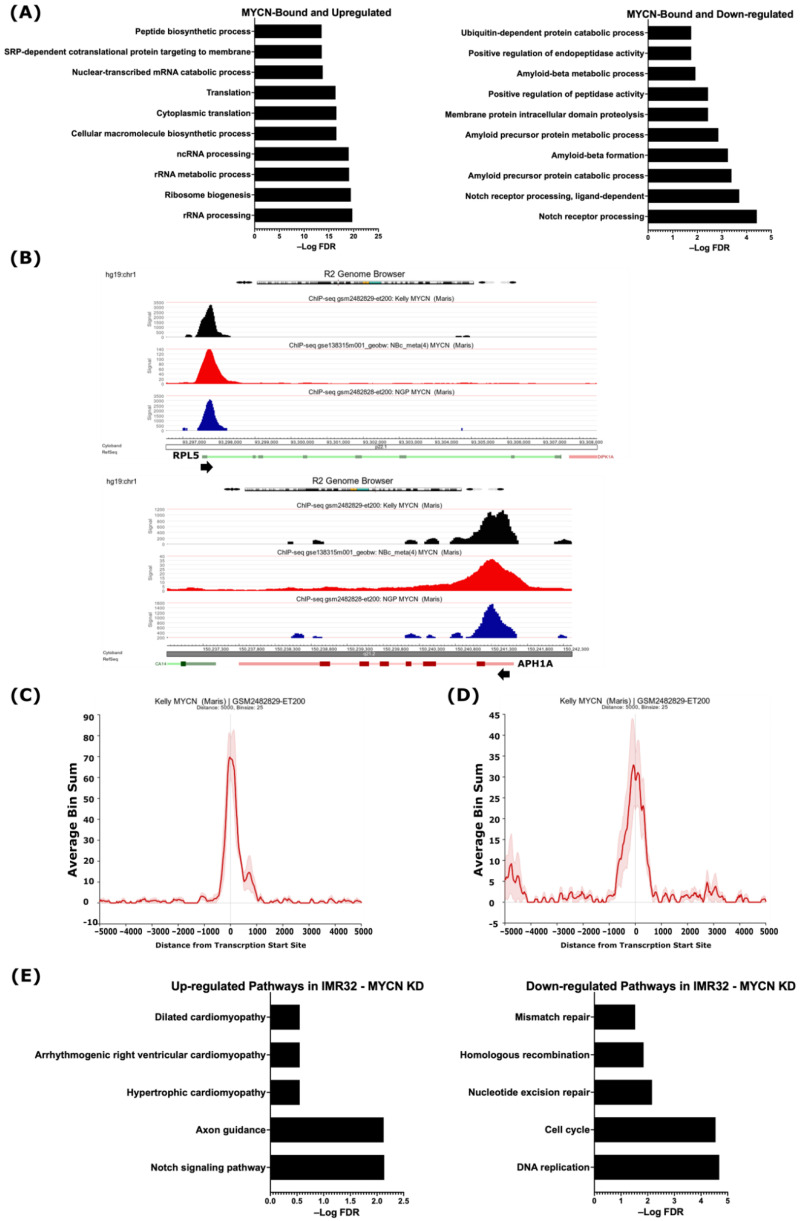
MYCN is a potential direct regulator of genes involved in differentially regulated biological processes. (**A**) Top 10 gene ontologies enriched with MYCN-bound genes. (**Left**), up-regulated genes. (**Right**), down-regulated genes. (**B**) R2 genome browser analysis of MYCN ChIP-Seq peaks at TSSs of the *RPL5* gene, representative of MYCN-bound and up-regulated genes involved in ribosomal biogenesis (**Top** panel), and the *APH1A* gene, representative of MYCN-bound and down-regulated genes involved in NOTCH signaling pathway (**Bottom** panel). Arrowhead indicates the direction of transcription. MYCN ChIP-Seq pileup at the TSSs of genes involved in (**C**) ribosomal biogenesis and (**D**) NOTCH signaling pathway. (**E**) Pathway analysis of differentially up-regulated (**left**) and down-regulated (**right**) genes in IMR32 cell line following *MYCN* knockdown. False discovery rate (FDR) ≤ 0.05 was used as cut-off value to define statistical significance.

**Figure 3 ijms-24-08141-f003:**
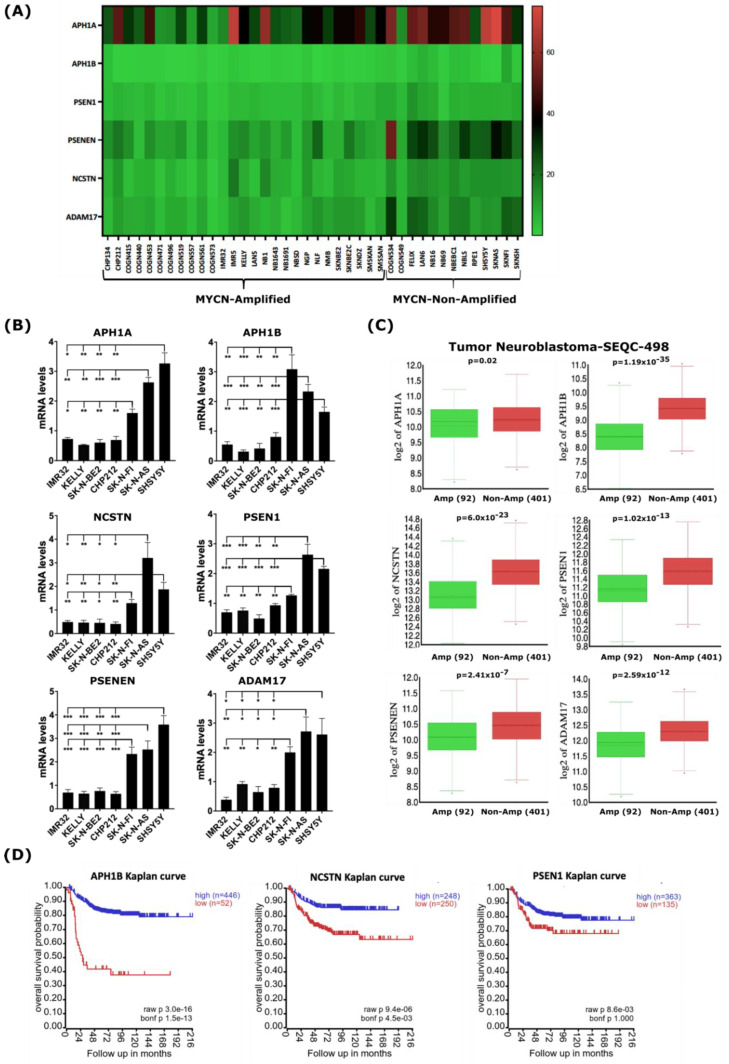
*MYCN* amplification correlates with reduced expression of genes encoding γ-secretase subunits and ADAM17 in human NB cell lines and primary tumors. (**A**) Heatmap analysis of mRNA levels of the γ-secretase complex genes and *ADAM17* in the 39-human NB-cell-line dataset. (**B**) RT-qPCR analysis of baseline expression levels of genes of the γ-secretase complex and *ADAM17* in a set of human NB cell lines. Error bars in panel represent the standard deviation of three independent biological experiments. *p*-value was calculated using a two-tailed, unpaired student *t*-test. * *p*-value ≤ 0.05; ** *p*-value ≤ 0.01; *** *p*-value ≤ 0.001. (**C**) In silico analysis of mRNA levels of γ-secretase complex genes and *ADAM17* in primary NB tumors with and without *MYCN* amplification. Statistical analysis was performed using one-way ANOVA. (**D**) Kaplan–Meier-analysis overall survival curve from the Tumor Neuroblastoma-SEQC-498 study based on mRNA expressions of genes of the γ-secretase complex, *APH1B*, *NCSTN*, and *PSEN1*. Statistical analysis was performed using Bonferroni correction of raw *p*-values.

**Figure 4 ijms-24-08141-f004:**
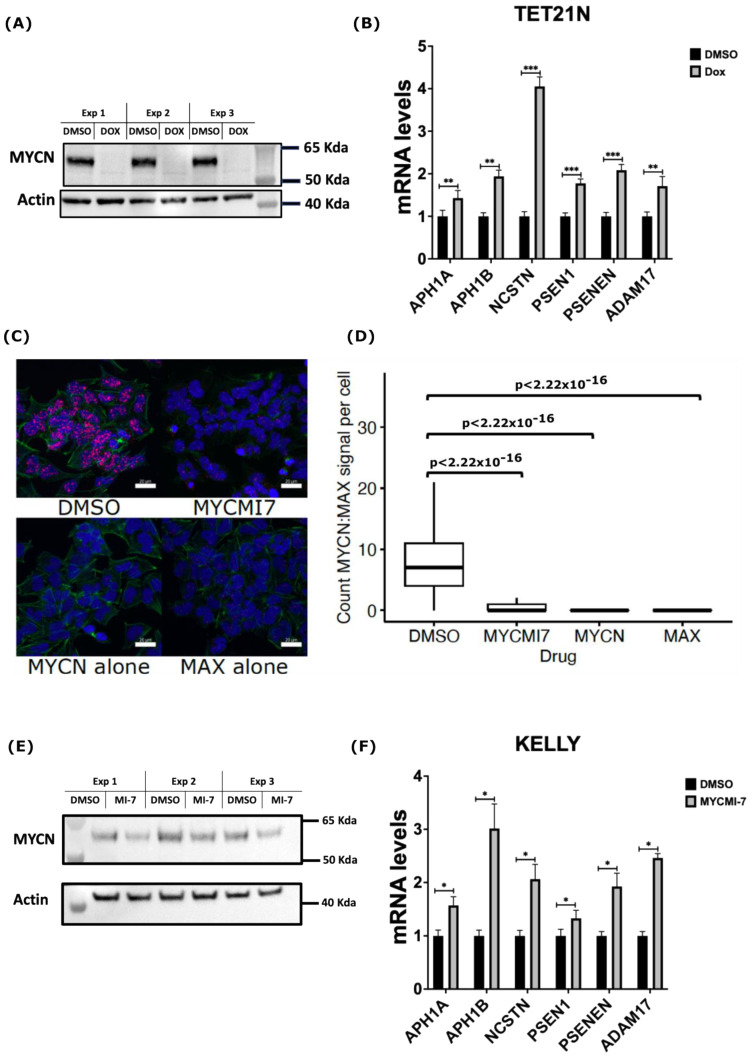
MYCN depletion induces expression of genes encoding γ-secretase members and ADAM17. (**A**) Western blot analysis of MYCN protein levels following treatment of TET21N cell line with 1 µg/mL doxycycline (Dox) for 72 h. (**B**) RT-qPCR analysis of mRNA levels of genes encoding γ-secretase complex and ADAM17 in response to MYCN Dox-induced depletion in the TET21N cell line. (**C**,**D**) Detection of endogenous MYCN:MAX interactions by isPLA in the KELLY cell line. Treatment with 5 µM of MYCMI-7 for 5 h disrupts MYCN:MAX interactions. (**C**) Representative images of MYCN:MAX isPLA signals in the KELLY cell line. Scale bar is 20 µM. (**D**) Quantification of MYCN:MAX isPLA signals in the KELLY cell line. isPLA signals using MYCN or MAX antibodies alone were used as background levels. DAPI was used to stain the nucleus (Blue), Phalloidin (AF488, Green) was used to stain the cytoplasm, and isPLA signals (Atto 647N, Red) detected MYCN-MAX interactions. (**E**) Western blot analysis of MYCN protein levels in the KELLY cell line following 72 h treatment with 0.4 µM MYCMI-7. (**F**) RT-qPCR analysis of mRNA levels of genes encoding the γ-secretase complex and ADAM17 in response to MYCN pharmacological depletion in the KELLY cell line. Error bars represent the standard deviation of three independent biological experiments. For the qPCR, the *p*-value was calculated using two-tailed, unpaired Student *t*-test. * *p*-value ≤ 0.05; ** *p*-value ≤ 0.01; *** *p*-value ≤ 0.001. For the isPLA, the statistical significance was calculated using ANOVA in R.

**Figure 5 ijms-24-08141-f005:**
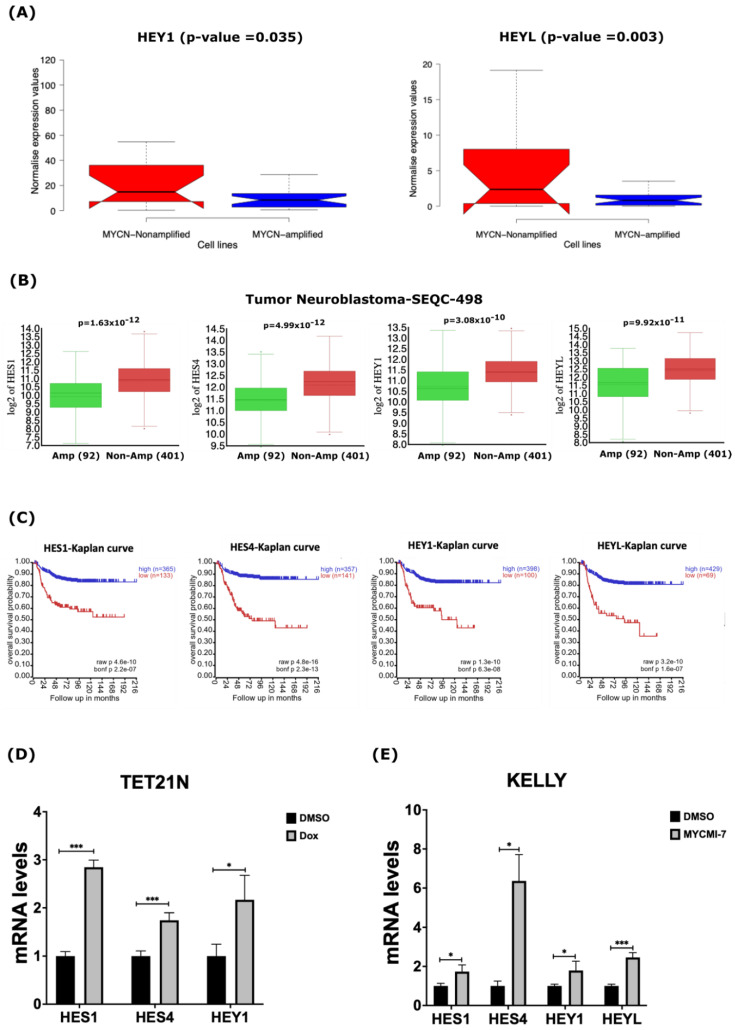
MYCN depletion induces expression of NOTCH target genes. (**A**) In silico expression analysis of *HEY1* and *HEYL* NOTCH target genes in 39 human NB cell lines. (**B**) In silico analysis of mRNA levels of NOTCH target genes *HES1*, *HES4*, *HEY1*, and *HEYL* in primary NB tumors. (**C**) Kaplan–Meier-analysis overall survival curve from the Tumor Neuroblastoma-SEQC-498 study based on mRNA expression of NOTCH target genes *HES1*, *HES4*, *HEY1*, and *HEYL*. (**D**,**E**) Expression of HES1, HES4, HEY1, and HEYL in response to 1 µg/mL doxycycline in (**D**) TET21N cell line or (**E**) KELLY cell line following 0.4 µM MYCMI-7 treatment. Cell lines were treated with doxycycline or MYCMI-7 for 72 h. DMSO was used as control treatment. Statistical significance was calculated using (**A**) unpaired Student *t*-test in panel, and (**B**) one-way ANOVA in panel. (**C**) Statistical analysis was performed using Bonferroni correction of raw *p*-values available in panel. For qPCR, *p*-value was calculated using two-tailed, unpaired Student *t*-test. * *p*-value ≤ 0.05; *** *p*-value ≤ 0.001.

**Figure 6 ijms-24-08141-f006:**
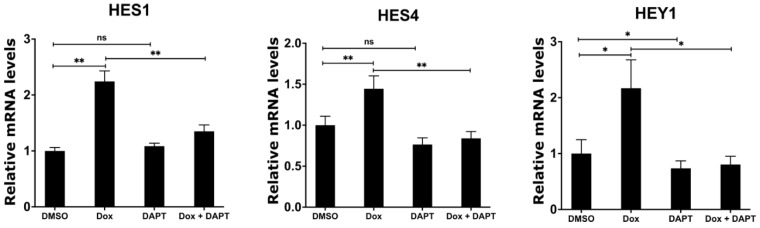
Inhibition of γ-secretase complex counteracts the induced expression of NOTCH target genes upon MYCN depletion. TET21N cell line was treated with 1 µg/mL of doxycycline, 5 µM of DAPT, or a combination for 72 h. DMSO treatment was used as control. Error bars represent the standard deviation of three independent biological experiments. *p*-value was calculated using two-tailed, unpaired Student *t*-test. * *p*-value ≤ 0.05; ** *p*-value ≤ 0.01; ns: no significance.

**Table 1 ijms-24-08141-t001:** Description of the NOTCH receptor processing enzymes identified in this study.

Gene Symbol	Gene Name	Function
*APH1A*	Aph-1 Homolog A	Promotes nicastrin and presenilin association in the γ-secretase complex, and mediates γ-secretase interaction with substrates prior to cleavage
*APH1B*	Aph-1 Homolog B	Presenilin-stabilizing cofactor
*NCSTN*	Nicastrin	Cleaves integral membrane proteins, including NOTCH receptors. It is believed to function as a stabilizing cofactor for gamma-secretase complex assembly
*PSEN1*	Presenilin 1	Protease, regulates gamma-secretase activity, cleaves the intracellular domain of NOTCH receptors leading to the production of active NOTCH intracellular domain (NICD)
*PSENEN*	Presenilin enhancer	Protease, regulates gamma-secretase activity, cleaves the intracellular domain of NOTCH receptors, leading to the production of active NICD
*ADAM17*	Disintegrin and metalloprotease domain 17	Membrane-anchored protease, mediates the initial cleavage of the extracellular domain of NOTCH receptors

All information related to the functions of these proteins were retrieved from the NCBI.

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
