# Peer review of "MYCN Amplification Is Associated with Reduced Expression of Genes Encoding γ-Secretase Complex and NOTCH Signaling Components in Neuroblastoma"

_ijms, 2023, doi:10.3390/ijms24098141_

Round 1

Reviewer 1 Report

In this manuscript, Alzigrat et al. seek to determine the transcriptional effects of MYCN overexpression in neuroblastomas (NB) using mostly computational approaches to quantify expression in NB models and public primary tumor data.  The authors argue that MYCN is directly responsible for activating and repressing expression of many genes with functions ranging from protein synthesis to intercellular signaling.  The therapeutic challenges presented by MYCN overexpression are substantial, as it is one of the strongest correlates with poor outcomes for NB patients, so the work is of interest.  The techniques used are generally reasonable, but have specific limitations I describe below.  These limitations call some of the overall conclusions of the manuscript into question and temper my enthusiasm about the work.

MYCN has been noted, like its family member c-MYC, to act as a transcriptional amplifier (PMID: 20434984, 29379199). That is, it is mechanistically responsible for activating transcription elongation from virtually every expressed gene. Because of the assumptions made in bulk RNA-seq analysis, exogenous normalization measures that account for per-cell mRNA production are necessary to correctly understand the results of altering MYC gene expression (PMID: 23101621).  

Major Points:
1) The vast majority of publicly available RNA-seq data do not account for per-cell mRNA amount differences. This makes it difficult to believe some of the author's conclusions about MYCN-regulated genes.  It is expected that higher MYCN levels result in higher amounts of mRNA being produced overall.  I believe the author's figures that there are differences in the relative amounts of mRNA per cell for genes in the different gene ontology terms, but calling them "upregulated" or "downregulated" is misleading at best.  This is also complicated by observations that c-MYC often substitutes for MYCN in some of these same NB lines (PMID: 29284669).  Terms should be substituted for "more/less highly expressed" or clearly specifying that these are relative values, which account for this ambiguity.

2) I strongly doubt that there are genes that are directly repressed by MYCN, based on the mechanistic observations from publications mentioned above.  Expressed genes are expected to have MYCN bound at their promoters in MYCN-expressing cells.  Loss of MYCN from a system is expected to have globally suppressing effects on transcription elongation.  RT-qPCR also does not, as a default, account for per-cell mRNA production differences.  At several days without MYCN, it would be expected that NB cells are dying.  The authors do not control for cell death.  The authors also do not account for global transcriptional shifts, i.e. global loss of expression, when MYCN is lost.  The authors also observe that Actin is substantially and consistently negatively affected by MYCMI-7, consistent with a global loss of expression.  To argue for direct MYCN repression, the authors would need to perform spike-in normalized mRNA-seq to account for MYCN-based amplification changes. An assessment of cell proliferation under the treatment regime is needed.  Immunofluorescence analysis of NOTCH family members might be beneficial, as the authors have a reason to be interested in this pathway and its members.

Minor Points:
1) When the authors refer to "amplified," do they consider gene amplification as the same consequence as some other form of over-expression of MYCN?

2) Gene orientation is not noted in the Genome Browser tracks, so the end representing the promoter is only able to be guessed at.

3) There are grammar and spelling issues throughout.

Reviewer 2 Report

2210621

Analysis of MYCN ChIP-Seq revealed enrichment of MYCN binding at the transcription start site of genes encoding γ-secretase complex subunits. 

we revealed that the expression of γ-secretase subunits encoding genes and other components of the NOTCH pathway was also reduced in MYCN-amplified tumors and correlated with worse overall survival in NB patients. Does the expression of any of the Notch signalling pathway members correlate with favourable prognosis in these patients?

Amplification of MYCN gene is detected in ∼20% of all NB cases and about 40% of high-risk cases. It is considered the genetic aberration most consistently associated with high-risk disease and poor survival. This is good but the authors could also talk about TERT and ALT phenotypes, as other cases of high-risk NB.

The MYC:MAX complexes can either activate or repress gene expression through MYC’s ability to recruit different cofactors including chromatin modifiers allowing or preventing the transcription of the corresponding target gene/s in a context-dependent manner. This mechanism described is standard for all transcription factors with activator or inhibitor roles, what other specific detail can be provided about MYC? Do the authors have any specific information to add about MYCN as a member of the MYC family, and its specific transcriptional regulatory roles in neuroblastoma?

We uncover that in MYCN-amplified cell lines upregulated genes were associated with ribosome biogenesis, RNA metabolism, gene expression, protein synthesis processes and down-regulated genes were mainly associated with NOTCH-signaling, axoneme assembly, protein localization and transport. This may be the case since MYCN-amplified cell lines will be very diverse, and it may be difficult to assign specific gene expression patterns to MYCN amplification only, it may be good to specifically manipulate MYCN and then test the altered gene expression and compare it to the data described above to verify.

Methods:

In lines 440-447, the authors should provide detail about how the data were normalised the RNA-seq data reads, for example, detail about raw read normalisation, perhaps if the data was from various sources and had different read lengths, how was this accounted for? Also, what tools were used for mapping to the transcriptome and calling differentiation expression (i.e., Salmon, RNA-Star and DESeq2)? This type of information is essential and has not been provided.

Figure 1. MYCN-differentially regulated pathways in NB. Top 10 differentially upregulated (A) and downregulated (B) pathways in human NB MYCN-amplified cell lines when compared to MYCN-non-amplified cell lines. (C) Overlap between common MYCN target genes identified in three human MYCN-amplified cell lines and differentially regulated genes in MYCN-amplified human NB cell lines. As mentioned earlier, it is logical to compare two conditions that differ in one parameter that is shown figure 1A/B here, however, one concern is the diversity within the MYCN amplified and non-amplified neuroblastomas and perhaps grouping them together could lead to errors. The authors could show the differential expression of the 38 cell lines using hierarchical clustering and evaluate how different these expression patterns are and then perhaps decide if it is a better idea to class them together or not (or indeed with cell lines have similar expression patterns and may be classed into subgroups). Alternatively, MYCN could be specifically targeted to understand the resulting differential expression (this tool is available to the authors). 

Also, it could be an idea to specifically test the expression of the relevant NOTCH signalling pathway genes in this study in some of these 38 cell lines to look for any inter-MYCN amplified and inter-MYCN nonamplified variations. I realise that this has been done for a handful in figure 2 but what about the 38 cell line-worth of data that have contributed to the differential expression evidence but have not been individually shown or validated?

It is good that the authors have tried to at least link differential gene expression with MYCN genome-wide binding, but binding does not always lead to regulation, especially with the promiscuous MYC family which has a massive number of binding sites in the genome. Also, please show more evidence of MYCN binding to the regulatory regions of the Notch genes mentioned in this study, perhaps an extra figure could be added,

The topics discussed in lines 131- 140 are more like introductions than results.

In figure 1F, why were those genes selected?

In figure 2C, why have KM curves only been shown for the three of the genes, please expand.

Analysis of mRNA levels of ADAM17 and genes encoding the γ-secretase complex by RT-qPCR revealed a significant increase in mRNA levels following doxycycline treatment (Figure 3B). Since the authors have an inducible method for MYCN repression then perhaps some RNA-seq analyses or at least large-scale gene expression analyses could verify the differentially expressed genes derived from comparing MYCN-amplified/ non-amplified cell lines that will be heterogeneous.

In figure 3, the DOX system seems to be working quite well without any leaks. Also, the other data in figure 3 is convincing (both methods of repressing MYCN). Also, the data in figures 4-5 then test some Notch targets that are affected by MYCN using the same established systems. What is the expression pattern of gamma-secretase itself in the presence and absence of MYCN (genetic and pharmacological manipulation)? If gamma-secretase reverses the effect of inhibiting MYCN, then what is the relationship between it and MYCN? The abstract claims that MYCN suppresses gamma-secretase but this evidence needs to be shown in more detail.

Van Groningen and colleagues and also other neuroblastoma research groups have proposed the presence of epigenetic states forming core regulatory circuities including ADRN and MES in this cancer, the latter showing Notch receptors and cofactors such as NOTCH1/2/3, MAML2 and also Notch pathways target genes such as HES1. MES state, however, is linked to higher migration rates and therefore displays a more aggressive phenotype. How can this be linked to this study in which the MYCN depletion increases the levels of Notch pathway genes, while the expression of Notch pathway members, based on these epigenetic/super-enhancer studies, is linked to more aggressive neuroblastoma states? 

What is the implication of this study from a therapeutic angle? It is a difficult question I think since both MYCN and Notch signalling pathways are extremely multifucntional and their manipulation may indeed be tricky and context-dependent. 

Round 2

Reviewer 2 Report

The authors have addressed my comments.